# Overall Survival in Real-World Patients with Unresectable Hepatocellular Carcinoma Receiving Atezolizumab Plus Bevacizumab Versus Sorafenib or Lenvatinib as First-Line Therapy: Findings from the National Veterans Health Administration Database

**DOI:** 10.3390/cancers16203508

**Published:** 2024-10-17

**Authors:** David E. Kaplan, Ruoding Tan, Cheryl Xiang, Fan Mu, Sairy Hernandez, Sarika Ogale, Jiayang Li, Yilu Lin, Lizheng Shi, Amit G. Singal

**Affiliations:** 1Department of Medicine, Perelman School of Medicine, University of Pennsylvania, Philadelphia, PA 19104, USA; 2Gastroenterology Section, Corporal Michael J. Crescenz VA Medical Center, Philadelphia, PA 19104, USA; 3Genentech, South San Francisco, CA 94080, USA; tan.ruoding@gene.com (R.T.); hernandez.sairy@gene.com (S.H.); ogale.sarika@gene.com (S.O.); 4Analysis Group Inc., Boston, MA 02199, USA; cheryl.xiang@analysisgroup.com (C.X.); fan.mu@analysisgroup.com (F.M.); jiayang.li@analysisgroup.com (J.L.); 5Department of Health Policy and Management, Tulane University, New Orleans, LA 70118, USA; ylin15@tulane.edu (Y.L.); lshi1@tulane.edu (L.S.); 6New Orleans VA Medical Center, New Orleans, LA 70119, USA; 7Department of Internal Medicine, University of Texas Southwestern Medical Center, Dallas, TX 75390, USA; amit.singal@utsouthwestern.edu

**Keywords:** atezolizumab, bevacizumab, hepatocellular carcinoma, lenvatinib, National Veterans Health Administration database, overall survival, real world, sorafenib

## Abstract

Hepatocellular carcinoma is the most common type of liver cancer and is often seen in people with chronic liver diseases. For patients with hepatocellular carcinoma who cannot receive surgery, other treatments include atezolizumab plus bevacizumab, sorafenib, or lenvatinib. We compared the length of life among 1874 United States veterans who received these therapies, both in the overall population of patients and in groups by race/ethnicity. The results showed that the risk of death was 30% lower with atezolizumab plus bevacizumab compared with sorafenib, and 26% lower compared with lenvatinib. These trends were consistent across White, Black, and Hispanic patients. Thus, patients who received atezolizumab plus bevacizumab had improved survival outcomes compared with those treated with sorafenib or lenvatinib, regardless of race or ethnicity.

## 1. Introduction

Hepatocellular carcinoma (HCC) is the most common form of liver cancer in adults and the third leading cause of cancer deaths globally [1,2]. HCC commonly develops in the presence of cirrhosis although additional risk factors include hepatitis B or C virus infection, alcohol use, diabetes, and obesity across various populations [3,4]. The overall prognosis of HCC depends on tumor stage and liver dysfunction, and the five-year overall survival (OS) of patients with unresectable HCC (uHCC) is just 12% [5].

Until recently, the tyrosine kinase inhibitors (TKIs) sorafenib and lenvatinib were widely used as the first-line (1L) therapy for patients with uHCC [6]. These TKIs were associated with similar OS and safety profiles as 1L therapy for uHCC liver [7]. However, the landscape of HCC treatment evolved following the US Food and Drug Administration (FDA) approval of immunotherapies for uHCC, beginning in May 2020 with the approval of atezolizumab plus bevacizumab as a 1L treatment for adults with unresectable or metastatic HCC [8]. Atezolizumab, an immune checkpoint inhibitor, is a monoclonal antibody that binds to programmed death ligand 1 (PDL1) and blocks its interaction with PD1 and B7.1 receptors [8]. Bevacizumab is a humanized monoclonal IgG1 antibody that binds to vascular endothelial growth factor (VEGF) ligands and has anti-angiogenic and immuno-modulatory effects [9]. Clinical and preclinical studies demonstrated that anti-PD-L1 plus anti-VEGF lead to a significant improvement in survival and enhanced anti-tumor activity [10]. The National Comprehensive Cancer Network^®^ (NCCN^®^) recommended this combination as an NCCN Category 1 preferred systemic treatment option for certain patients with 1L uHCC [11]. This approval was based on the results of the Phase III IMbrave150 clinical trial, which observed a 42% reduction in mortality risk compared with sorafenib [12]. There are no head-to-head clinical trials comparing the efficacy of atezolizumab plus bevacizumab with that of lenvatinib, although the REFLECT trial found lenvatinib to be non-inferior in efficacy to sorafenib [13].

There is limited real-world evidence comparing the effectiveness of atezolizumab plus bevacizumab with that of TKIs among patients in the US as most existing real-world studies have been conducted outside of the US. Due to the complex pathophysiology involved in HCC development, several efforts have been made to elucidate the impact of HCC etiology on treatment response and survival outcomes. For example, some prior real-world studies have observed an association between nonviral etiology and worse prognosis in patients treated with immune checkpoint inhibitors [14], likely due to a different hepatic microenvironment that consequently affects treatment response. However, the evidence for causality remains inconclusive [15].

Importantly, racial and ethnic disparities in HCC incidence, presentation, treatment accessibility, and outcomes present unique challenges in the US real-world practice [16]. Racial and ethnic minorities are disproportionately burdened by higher incidences of HCC and more often present more advanced disease stages at diagnosis but have reduced treatment rates than nonminority counterparts, contributing to worse survival rates [17,18,19]. In addition, the varying prognosis by race and ethnicity and the variability in chronic liver disease etiology across these groups may further influence their response to immunotherapy (IO) [20,21]. Last, racial and ethnic minorities have been notably underrepresented in clinical trials for uHCC, resulting in limited clinical evidence on racial diversity to date [22]. Thus, there is a critical need for inclusive research reflecting racial and ethnic diversity in treatment outcomes. Rigorous real-world evidence accounting for patient differences, particularly focusing on patients in the US, is warranted to compare the effectiveness of 1L atezolizumab plus bevacizumab vs. sorafenib or lenvatinib in uHCC. To bridge these evidence gaps, this study utilized the National Veteran Health Administration (VHA) database, which represents the largest healthcare system providing liver care in the US. The study described and compared patient characteristics and OS between patients with uHCC treated with atezolizumab plus bevacizumab vs. sorafenib or lenvatinib as 1L treatment. Additionally, this study assessed how the relative effectiveness of these treatments differed across racial and ethnic groups.

## 2. Methods

### 2.1. Data Source

This population-based retrospective cohort analysis used data from electronic medical records from the VHA Data Warehouse, spanning from 1 January 2017 to 31 December 2022. The VHA is the largest integrated healthcare system in the US, providing comprehensive care to over 9 million veterans and their dependents annually, and is the nation’s leading provider of liver-related healthcare.

This study received the Institutional Review Board approval from the Southeast Louisiana Veterans Health Care System (SLVHCS).

### 2.2. Study Population

The study population consisted of patients diagnosed with HCC (International Classification of Diseases, 10th edition [ICD-10] code C22.0 and C22.8) who initiated 1L treatment with atezolizumab plus bevacizumab, sorafenib, or lenvatinib for unresectable or metastatic disease. The index date was defined as the 1L treatment initiation date. To ensure that patients had uHCC, additional exclusion criteria included the presence of other primary cancers (except for prostate, bladder, and non-melanoma malignant neoplasm of the skin) within 5 years of the index date, liver transplant prior to the index date, and liver surgery within 6 months (to exclude possible adjuvant therapy) before the index date. To identify 1L treatment, additional exclusion criteria included the use of systemic treatments after the initial HCC diagnosis and prior to the index date, clinical trial participation on or after the index date, and surgery any time after the index date (to exclude possible downstaging).

Among eligible patients with uHCC, sub-cohorts were constructed to include White (non-Hispanic), Black (non-Hispanic), and Hispanic patients for the analyses by race and ethnicity.

### 2.3. Study Variables

Patient demographic (i.e., age, sex, race and ethnicity, region) and clinical characteristics were collected during the one-year baseline period prior to the index date. Race and ethnicity data were self-reported by patients during their interactions with the VA healthcare system. Clinical characteristics included liver disease etiology, presence of cirrhosis, Charlson Comorbidity Index (CCI) score, presence of other comorbidities (diabetes, hypertension), Child–Pugh score, modified albumin–bilirubin (mALBI) grades, prior local therapies, and extrahepatic metastases. Specifically, the Child–Pugh score was derived using a validated algorithm developed within the VHA data, which had a high accuracy rate of 96–98% [23]. This algorithm quantified the severity of ascites and encephalopathy by evaluating complications and treatment, and established guidelines for identifying the most representative laboratory data. Similarly, the mALBI grades were calculated based on serum albumin and bilirubin levels following a validated scoring system [24]. Missing laboratory data were imputed based on the validated Child–Pugh algorithm with the last value carried forward method.

OS was recorded for each cohort, defined as the interval between the start of 1L treatment and death from any cause. Patients who were still alive were censored at the end of the VA enrollment or the study end date, whichever occurred first.

### 2.4. Statistical Analysis

Patient characteristics during the baseline period were compared across index treatments using Wilcoxon rank sum tests for continuous variables and chi-square tests for categorical variables. Unadjusted OS was calculated using the Kaplan–Meier (KM) method, and the probability of survival at a given time point was derived directly from the KM survival curve. A comparison of unadjusted OS curves by cohort was conducted using the log-rank test. A multivariate Cox proportional hazard regression model, adjusted for age, race, region, liver disease etiology, presence of cirrhosis, CCI score, Child–Pugh score, mALBI grade, prior local therapies, and extrahepatic metastases, was applied to compare the effectiveness of atezolizumab plus bevacizumab vs. sorafenib and vs. lenvatinib. These covariates were selected based on clinical input and significant differences in patient characteristics observed across treatment cohorts. Hazard ratios (HR) and corresponding 95% confidence intervals (CI) were reported. An assessment of all Cox proportional hazards model assumptions was conducted.

Given the prior evidence in the literature of racial and ethnic disparities in survival outcomes and differences in treatment responses, race/ethnicity was evaluated as a treatment effect modifier for HCC. Thus, the multivariate Cox regression model was augmented by including the interaction terms between the index treatment and race/ethnicity to assess how OS differed between patients treated with 1L atezolizumab plus bevacizumab vs. TKIs across racial and ethnic groups. Additional exploratory descriptive KM analyses of OS were conducted in subgroups stratified by Child–Pugh score (class A [5–6 points] and class B [7–9 points]), mALBI grade (grade 1, 2A, and 2B), and liver disease etiology (viral vs. nonviral). Statistical significance was considered at a level of 0.05 in all analyses.

### 2.5. Software

Analyses were conducted using SAS 9.4 software (SAS Institute Inc., Cary, NC, USA) and R software with the survival package (version 3.4.2; R Foundation for Statistical Computing, Vienna, Austria).

## 3. Results

### 3.1. Sample Selection

A total of 1874 patients of any race and meeting all study criteria were identified in the VHA data (overall population). Among these patients, 405 (21.6%) received atezolizumab plus bevacizumab, 1016 (54.2%) received sorafenib, and 453 (24.2%) received lenvatinib as 1L treatment for uHCC (Figure 1).

### 3.2. Overall Population of Patients with uHCC

#### 3.2.1. Comparison of Patient Characteristics Across Treatment Groups

Patient baseline demographics were comparable across treatment groups in the overall population (Table 1). The mean age was approximately 69 years (mean age range: 69.3–69.5 years) and almost all patients were male (proportion range: 98.0–99.3%).

Across the three treatment groups, there were similar proportions of patients with viral etiology (proportion range: 64.2–67.7%), nonviral etiology (11.9–12.8%), and cirrhosis (69.3–73.1%) (all *p* > 0.05). Compared with patients receiving 1L sorafenib or lenvatinib, significantly higher proportions of patients initiating 1L atezolizumab plus bevacizumab had extrahepatic metastases (28.1% vs. 14.5% [sorafenib] and 19.0% [lenvatinib]; both *p* < 0.01). Patients treated with 1L atezolizumab plus bevacizumab, compared with those receiving sorafenib or lenvatinib, had a higher proportion of Child–Pugh A (85.2% vs. 67.0% [sorafenib] and 76.8% [lenvatinib], respectively) and a lower proportion of Child–Pugh B (13.8% vs. 31.4% and 22.3%), indicating a better liver function in the atezolizumab plus bevacizumab cohort (all *p* < 0.01). Specifically, patients treated with 1L atezolizumab plus bevacizumab had a higher proportion of Child–Pugh A5 (54.3% vs. 35.0% and 47.2%; both *p* < 0.05), a similar proportion of A6 (both *p* > 0.05), and a lower proportion of Child–Pugh B7 (9.1% vs. 19.6 and 15.5%; both *p* < 0.01) compared with those treated with sorafenib or lenvatinib. Furthermore, among patients treated with 1L atezolizumab plus bevacizumab, a higher proportion were classified as mALBI grades 1 and 2A compared with those receiving sorafenib (34.1% vs. 21.0% for grade 1, 34.6% vs. 27.5% for grade 2A; both *p* < 0.01), a similar proportion were classified as mALBI grades 1 and 2A compared with those receiving lenvatinib (34.1% vs. 31.6% for grade 1, 34.6% vs. 28.7% for grade 2A; both *p* > 0.05), and a lower proportion were classified as ALBI grade 2B compared with those receiving either TKI (26.2% vs. 39.5% [sorafenib] and 32.7% [lenvatinib]; both *p* < 0.05). A lower proportion of patients receiving 1L atezolizumab and bevacizumab had ascites compared with those treated with 1L sorafenib (6.2% vs. 15.1%; *p* < 0.001). Patients treated with 1L atezolizumab plus bevacizumab had a higher CCI score compared with patients receiving sorafenib or lenvatinib (mean 6.6 vs. 5.8 and 5.8; both *p* < 0.001).

#### 3.2.2. Difference in Overall Survival by Treatment Groups

The median duration of follow-up was 8.5 months for the atezolizumab plus bevacizumab cohort, 7.6 months for the sorafenib cohort, and 8.2 months for the lenvatinib cohort. Patients treated with atezolizumab plus bevacizumab had an unadjusted median OS of 12.8 (95% CI: 10.6, 17.1) months, which was significantly longer than for those treated with sorafenib (8.0 [7.1, 8.6] months) or lenvatinib (9.5 [7.8, 11.4] months; both log-rank *p* < 0.001) (Figure 2). The results of the adjusted Cox regression model showed that patients treated with atezolizumab plus bevacizumab had a 30% reduced risk of death in comparison to sorafenib-treated patients (adjusted HR: 0.70 [95% CI: 0.60, 0.82]) and a 26% reduced risk of death compared with lenvatinib-treated patients (0.74 [0.62, 0.88]; both *p* < 0.001).

### 3.3. Analysis by Race and Ethnicity

A total of 1738 patients were included in the analysis by race and ethnicity, including 1140 (65.6%) White, 437 (25.1%) Black, and 161 (9.3%) Hispanic patients. We excluded 136 patients with unknown race (n = 98) and other races, including Native American and Asian (n = 38). The distribution of index treatments by race and ethnicity is reported in Figure 1, with 376 (21.6%) being treated with atezolizumab plus bevacizumab, 943 (54.3%) with sorafenib, and 419 (24.1%) with lenvatinib. In comparison with White patients, a significantly higher proportion of Black patients were treated with atezolizumab plus bevacizumab (24.9% vs. 20.4%), while a significantly lower proportion was treated with sorafenib (49.7% vs. 55.6%; both *p* < 0.05) (Table 1).

There were several differences in clinical characteristics across the racial and ethnic groups (Table 1). Compared with White patients, Black patients had lower BMI (mean 26.4 vs. 28.3; *p* < 0.001), a higher proportion of patients with viral etiology (89.9% vs. 57.6%; *p* < 0.001), and a lower proportion with nonviral etiology (3.0% vs. 15.0%; *p* < 0.001). The distribution of Child–Pugh scores and mALBI grades were similar, although Black patients were less likely to have ascites than White patients (7.1 vs 13.6%; *p* < 0.001).

Survival in the cohorts of White, Black, and Hispanic patients was consistent with that observed in the overall population inclusive of all races and ethnicities. Specifically, the median OS among patients treated with atezolizumab plus bevacizumab was 13.3 [10.9, 17.4] months, which was significantly longer than for sorafenib-treated patients (7.9 [7.0, 8.5] months) or lenvatinib-treated patients (9.4 [7.8, 11.3] months; both log-rank *p* < 0.01). The results of the multivariable Cox models indicated that the risk of death was 31% (adjusted HR: 0.69 [95% CI: 0.59, 0.81]) and 26% (0.74 [0.62, 0.89]) lower in the atezolizumab plus bevacizumab group compared with the sorafenib and lenvatinib groups, respectively (both *p* < 0.001) (Figure 3A,B).

Across all racial and ethnic groups, patients treated with atezolizumab plus bevacizumab had a longer unadjusted median OS compared with those treated with sorafenib or lenvatinib (all log-rank *p* < 0.01; Figure 4A–C). The difference in unadjusted median OS was numerically greater in Black patients (median OS: 19.5 vs. 8.2 [sorafenib] and 11.5 months [lenvatinib]) and Hispanic patients (17.5 vs. 7.9 and 8.1 months, respectively) compared with White patients (11.1 vs. 7.6 and 8.7 months) (all log-rank *p* < 0.01). These results were consistent with those of the multivariate Cox regression, which showed that White, Black, and Hispanic patients treated with atezolizumab plus bevacizumab had a 27%, 35%, and 45% reduction in the risk of death, respectively, compared with those treated with sorafenib (all *p* < 0.05) (Figure 3A). White patients treated with atezolizumab plus bevacizumab also had a 23% reduction in the risk of death compared with patients treated with lenvatinib (p < 0.05) (Figure 3B). The risk of death among Black and Hispanic patients receiving atezolizumab plus bevacizumab was 30% and 33% lower vs. lenvatinib, respectively (Figure 3B). However, the results were not statistically significant (*p* = 0.05 and 0.20, respectively), possibly due to smaller sample sizes in these groups.

Although there was a trend of a numerically larger OS benefit with atezolizumab plus bevacizumab (compared with sorafenib and lenvatinib) among Black and Hispanic patients compared with White patients, the interaction terms between index treatment and race/ethnicity were not statistically significant after adjustment for disease etiology and other confounders (*p* >0.05 for all interaction terms).

### 3.4. Exploratory Subgroup Analysis by Liver Function and Etiology

Appendix A reports the unadjusted median OS in subgroups of patients stratified by liver function (Child–Pugh class and mALBI grade) and etiology (viral vs nonviral) across index treatments. Patients treated with atezolizumab plus bevacizumab who had better liver function (Child–Pugh A or mALBI grade 1 or 2A) or viral etiology appeared to exhibit longer OS compared with sorafenib- or lenvatinib-treated patients. OS did not appear to differ significantly across treatments among patients with poor liver function (Child–Pugh B or C, mALBI grade 2B) or nonviral etiology, as evidenced by overlapping 95% CIs.

## 4. Discussion

This retrospective cohort study of US patients in the VHA evaluated the survival benefit associated with atezolizumab plus bevacizumab compared with TKIs (sorafenib and lenvatinib) as a 1L treatment for uHCC. In the overall population, atezolizumab plus bevacizumab was associated with significantly longer OS compared with both sorafenib and lenvatinib. The unadjusted median OS for atezolizumab plus bevacizumab was 3–4 months longer than that of the TKIs and the risk of death was 26–30% lower. For context, the IMbrave150 trial reported a 42% reduction in the risk of death with atezolizumab plus bevacizumab compared with sorafenib in patients with Child–Pugh class A who met the other enrollment criteria [12]. The observed benefit was consistent across cohorts grouped by race and ethnicity. Taken together, these findings suggest that atezolizumab plus bevacizumab was associated with a meaningful survival advantage over TKIs, sorafenib and lenvatinib, in the overall patient population as well as across racial and ethnic groups.

To the best of our knowledge, this is one of the first US-based studies comparing the real-world effectiveness of atezolizumab plus bevacizumab vs. TKIs (sorafenib and lenvatinib) in uHCC. The real-world OS outcomes in the current study, with a median survival of 12.8 months for atezolizumab plus bevacizumab, are consistent with those of another recent VHA study reporting a median OS of 11.4 months for patients receiving 1L atezolizumab plus bevacizumab [25]. The present findings are also generally consistent with the results of similar observational studies conducted outside of the US, with some differences that could be attributed to variations in regional practice, health providers, and patient populations. For example, the unadjusted median OS of 12.8 months among patients receiving 1L atezolizumab and bevacizumab in the present study was similar to, but slightly lower than, the OS of 14.9 months reported in a global multi-center retrospective study of 216 atezolizumab plus bevacizumab patients with uHCC treated in seven countries, including the US [26]. A global retrospective study of 2205 patients comparing the effectiveness of 1L atezolizumab plus bevacizumab versus lenvatinib for uHCC identified a 24% reduction in the risk of death with atezolizumab plus bevacizumab compared with lenvatinib in patients with viral etiology [27]. This is similar to the findings in the present cohort (i.e., a 26% reduced risk of death), in which two-thirds of patients had viral etiology. A 2023 German study observed prolonged OS among real-world uHCC patients treated with atezolizumab plus bevacizumab vs. sorafenib or lenvatinib (20 vs. 10 months, *p* < 0.001) [28]. However, considering emerging data from clinical trials suggesting prolonged survival for lenvatinib, such as the LEAP-002 study [29], the observed survival benefit in this study should be considered alongside the data from clinical trials.

The current study also contributes to the understanding of racial and ethnic differences in comparative effectiveness of atezolizumab plus bevacizumab and TKIs. The OS benefit of atezolizumab plus bevacizumab over TKIs was evident in the racial and ethnic subgroups where the unadjusted median OS of 19.5 months for Black patients and 17.5 months for Hispanic patients treated with atezolizumab plus bevacizumab were approximately double that observed for patients treated with sorafenib or lenvatinib. Racial and ethnic variations in treatment response to immunotherapy, including atezolizumab plus bevacizumab, may arise from differences in genetic predispositions, access to care, or underlying disease etiologies [30,31]. Both race and geographical location are independent risk factors for HCC, while other risk factors (i.e., obesity, diabetes, hepatitis B and C, metabolic dysfunction-associated steatotic liver disease or steatohepatitis, smoking, etc.) are differentially prevalent among racial and ethnic subgroups [31]. Additionally, chronic stress related to poverty and/or discrimination, as well as lack of health insurance or difficulty accessing care may also impact cancer prognosis, treatment response, and timeliness of treatment initiation [31,32,33]. Further, there are racial and ethnic differences in health literacy and medical mistrust, which can impact engagement with medical care and adherence to treatment regimens [34]. Finally, racial differences in OS have been partially attributed to treatment delay and variation in utilization of HCC treatment [33,35]. The correlation between liver disease etiology and race is well established [17,20,31], which may influence the interpretation of results when evaluating race alone. However, after controlling for liver disease etiology as a confounder in the multivariate Cox regression, the OS benefit of atezolizumab plus bevacizumab persisted across racial subgroups with 23–45% reduced risk of death compared with TKIs among White, Black, and Hispanic patients. Notably, several studies demonstrated the underuse of treatment in racial and ethnic minorities, potentially related to a combination of structural and social barriers [19,34]. Our data highlight the importance of engaging disadvantaged patient populations with advanced-stage HCC to avoid losing the potential benefits of systemic therapies.

Exploratory subgroup analysis generated some additional data insights, suggesting that atezolizumab plus bevacizumab was associated with approximately 5–7 months longer median OS compared with TKIs in patients with less liver dysfunction (i.e., Child–Pugh score A, mALBI grade 1 and 2A) and approximately 5–8 months longer median OS in patients with viral etiology. Patients with nonviral etiology or those with more severe liver dysfunction had similar OS regardless of the type of 1L treatments. However, the subgroup analyses were exploratory, descriptive, and unadjusted. There have been several studies, including a post hoc analysis of the IMBrave150 study that showed no difference in immunotherapy efficacy by liver disease etiology [36]. Patients with greater liver dysfunction, particularly those with Child–Pugh C cirrhosis, have a high competing risk of mortality such that any differential benefits of systemic therapy are likely mitigated. Further, adverse events, including the risk of bleeding, are more common in these patients. This highlights the need for well-controlled studies such as the ongoing Phase 2 study assessing the safety and efficacy of atezolizumab plus bevacizumab in patients with Child–Pugh B7 and B8 liver cirrhosis (NCT06096779) [37]. Variations in patient characteristics and follow-up treatments across subgroups may impact differences in outcomes.

In addition to the limitations associated with the subgroup analysis, the results of this study are subject to limitations inherent to retrospective cohort analyses. First, there may be residual confounding as, despite similar demographics across patient cohorts and the inclusion of adjusted analyses, unobserved confounders could not be controlled and may introduce potential channeling bias in the interpretations of results. In the absence of a randomized controlled trial, the observational nature of this study does not allow for determinations of clinical superiority in these patients. Second, the present findings may not be fully generalizable to the wider uHCC population considering the demographic composition of the VHA dataset, which was almost totally composed of males (~99%). However, HCC is more prevalent in males than females, with a ratio of 4:1 [38,39]. Third, there is the potential for bias with regard to the classification for key variables such as etiology identified through diagnosis code and Child–Pugh class from a derived algorithm. Finally, there was a constrained sample size for Hispanic patients. Therefore, the results for this racial ethnic cohort should be interpreted with caution.

This US-based, real-world investigation benefits from multiple strengths, including the use of data from the VHA, one of the largest providers of liver-focused healthcare in the US. By leveraging this database, we captured a relatively large sample of patients who received atezolizumab plus bevacizumab and established robust relationships between the treatment choice and OS. Further, this study included diverse racial and ethnic groups, which enabled us to further understand how the OS benefit of atezolizumab plus bevacizumab over TKIs would manifest in the context of racial disparities. This study provides comprehensive, US-specific real-world evidence addressing a gap in the literature, which can be used by healthcare providers and patients during treatment decision-making. Further, the present results highlight the complex interaction of race, ethnicity, and clinical outcomes in uHCC treatment.

## 5. Conclusions

This real-world study demonstrated that atezolizumab plus bevacizumab was associated with a meaningful survival advantage over sorafenib and lenvatinib among patients overall and across racial and ethnic groups. While future research is warranted to further assess the comparative effectiveness of atezolizumab plus bevacizumab compared with TKIs among patient subgroups, this study provides valuable insights for guiding future trials and initiatives aimed at addressing racial disparities in HCC.

## Figures and Tables

**Figure 1 cancers-16-03508-f001:**
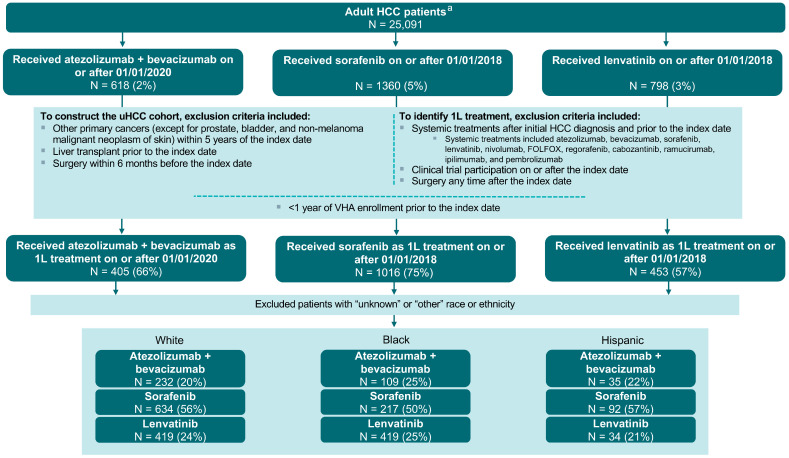
Sample selection flowchart of eligible patients with uHCC in the VHA data. Abbreviations: 1L, first-line; FOLFOX, 5-FU + leucovorin + oxaliplatin; HCC, hepatocellular carcinoma; uHCC, unresectable hepatocellular carcinoma; VHA, Veteran’s Health Administration. Notes: ^a^ Age ≥ 18 years at the index date.

**Figure 2 cancers-16-03508-f002:**
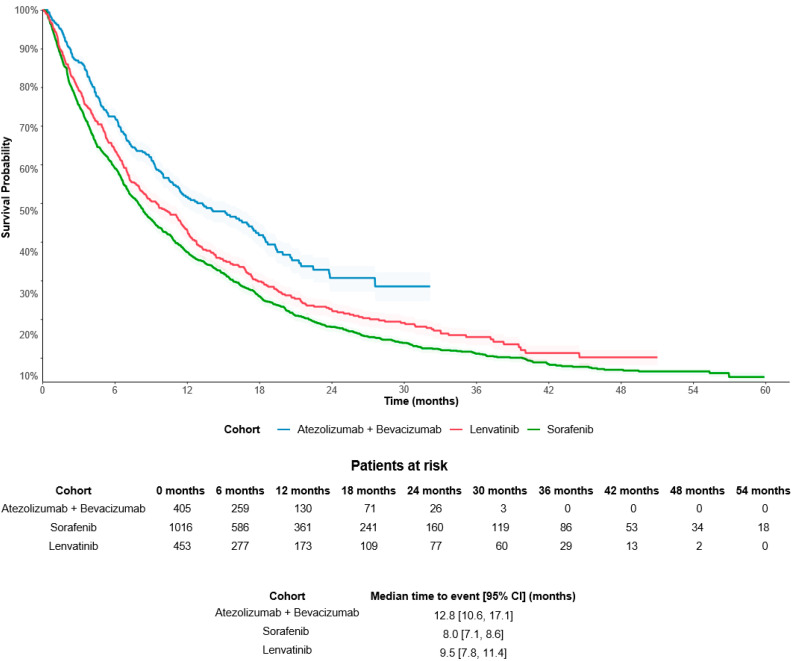
Unadjusted OS in the overall population of patients with uHCC. Abbreviations: 1L, first-line; HCC, hepatocellular carcinoma; OS, overall survival.

**Figure 3 cancers-16-03508-f003:**
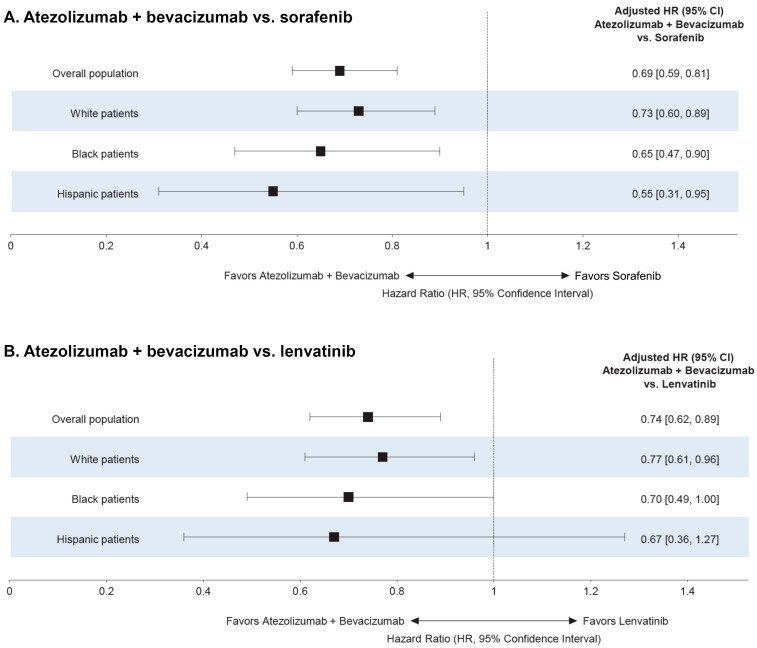
Adjusted HRs for OS comparing atezolizumab + bevacizumab vs. (**A**) sorafenib and (**B**) lenvatinib as a 1L treatment for uHCC in the analysis by race/ethnicity. In this figure, the combined race/ethnicity cohort refers to the cohort of patients included in the analysis by race/ethnicity (White, Black, and Hispanic patients). Abbreviations: 1L, first-line; CI, confidence interval; HCC, hepatocellular carcinoma; HR, hazard ratio; OS, overall survival.

**Figure 4 cancers-16-03508-f004:**
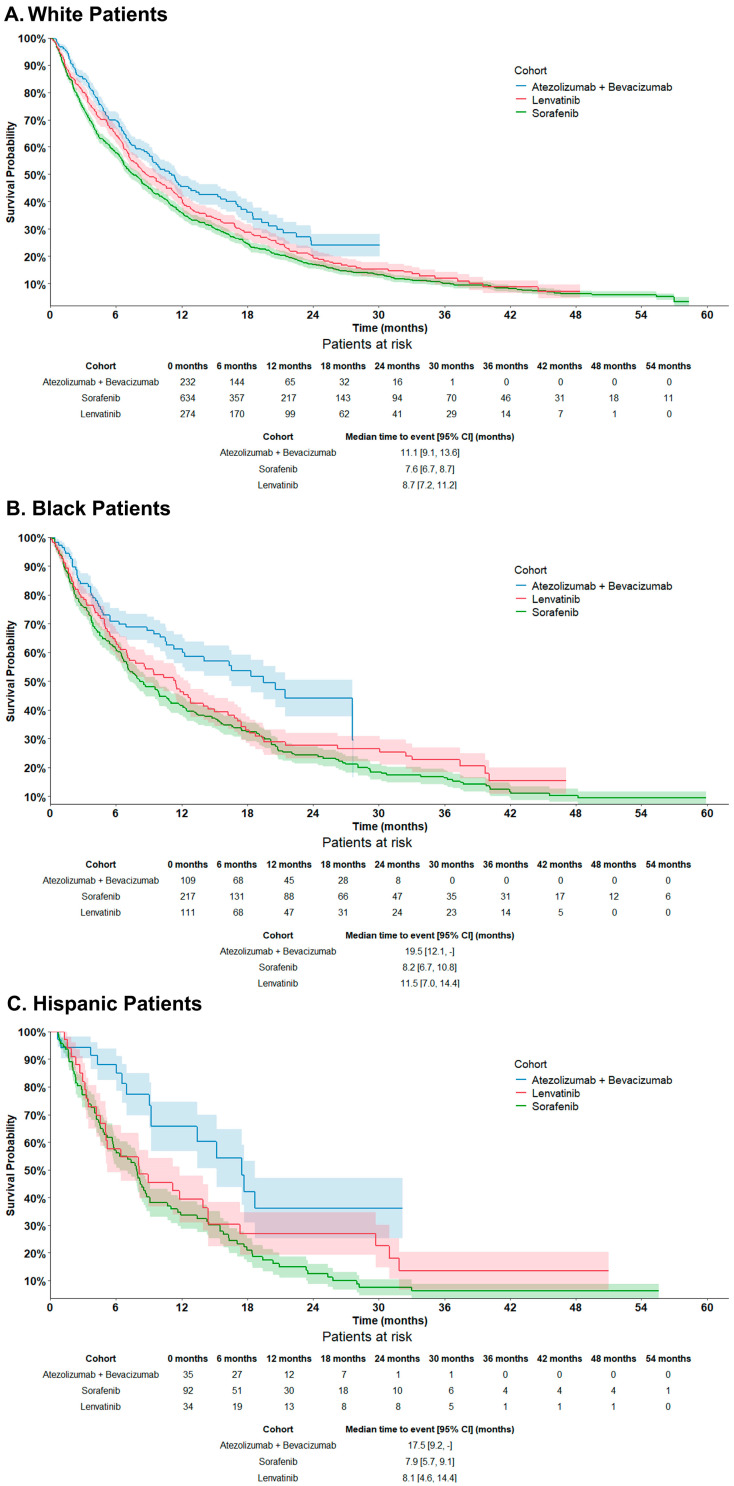
Unadjusted OS by index treatment among (**A**) White, (**B**) Black, and (**C**) Hispanic patients with uHCC. Abbreviations: 1L, first-line; CI, confidence interval; HCC, hepatocellular carcinoma; OS, overall survival.

**Table 1 cancers-16-03508-t001:** Demographic and baseline clinical characteristics among patients with uHCC, stratified by index treatment and race/ethnicity.

	By Index Treatment					By Race/Ethnicity				
	A + B	Sorafenib	A + B vs. Sorafenib *p*-Value	Lenvatinib	A + B vs. Lenvatinib *p*-Value	White	Black	White vs. Black *p*-Value	Hispanic	White vs. Hispanic *p*-Value
**Index treatment, n (%)**										
A + B	-	-	-	-	-	232 (20.4)	109 (24.9)	<0.05 *	35 (21.7)	0.683
Sorafenib	-	-	-	-	-	634 (55.6)	217 (49.7)	<0.05 *	92 (57.1)	0.715
Lenvatinib	-	-	-	-	-	274 (24.0)	111 (25.4)	0.572	34 (21.1)	0.415
**Race/ethnicity, n (%)**										
White, non-Hispanic	232 (57.3)	634 (62.4)	0.074	274 (60.5)	0.341	-	-	-	-	-
Black	109 (26.9)	217 (21.4)	<0.05 *	111 (24.5)	0.420	-	-	-	-	-
Hispanic	35 (8.6)	92 (9.1)	0.805	34 (7.5)	0.541	-	-	-	-	-
Native American	2 (0.5)	7 (0.7)	1.000	8 (1.8)	0.113	-	-	-	-	-
Asian	2 (0.5)	5 (0.5)	1.000	2 (0.4)	1.000	-	-	-	-	-
Other	4 (1.0)	5 (0.5)	0.285	3 (0.7)	0.713	-	-	-	-	-
Unknown	21 (5.2)	56 (5.5)	0.806	21 (4.6)	0.710	-	-	-	-	-
**Age in years, mean ± SD [median]**	69.5 ± 5.6 [69.6]	69.3 ± 6.4 [69.1]	0.559	69.3 ± 6.3 [69.3]	0.538	69.8 ± 6.2 [69.6]	68.1 ± 5.3 [68.0]	<0.001 ***	69.7 ± 7.6 [69.9]	0.808
**Male, n (%)**	397 (98.0)	1006 (99.0)	0.132	450 (99.3)	0.088	1128 (98.9)	429 (98.2)	0.217	161 (100.0)	0.381
**Geographic regions, n (%)**										
South	116 (28.6)	304 (29.9)	0.633	128 (28.3)	0.900	315 (27.6)	152 (34.8)	<0.01 **	44 (27.3)	0.936
Midwest	109 (26.9)	319 (31.4)	0.096	118 (26.0)	0.774	333 (29.2)	149 (34.1)	0.059	40 (24.8)	0.252
West	147 (36.3)	271 (26.7)	<0.001 ***	154 (34.0)	0.481	370 (32.5)	68 (15.6)	<0.001 ***	67 (41.6)	<0.05 *
Northeast	33 (8.1)	122 (12.0)	<0.05 *	53 (11.7)	0.084	122 (10.7)	68 (15.6)	<0.01 **	10 (6.2)	0.077
**Index year distribution, n (%)**										
2018	0 (0.0)	425 (41.8)	<0.001 ***	24 (5.3)	<0.001 ***	270 (23.7)	98 (22.4)	0.597	50 (31.1)	<0.05 *
2019	0 (0.0)	293 (28.8)	<0.001 ***	161 (35.5)	<0.001 ***	278 (24.4)	112 (25.6)	0.609	33 (20.5)	0.279
2020	86 (21.2)	169 (16.6)	<0.05 *	129 (28.5)	<0.05 *	244 (21.4)	84 (19.2)	0.339	32 (19.9)	0.657
2021	175 (43.2)	75 (7.4)	<0.001 ***	87 (19.2)	<0.001 ***	200 (17.5)	87 (19.9)	0.276	24 (14.9)	0.407
2022	144 (35.6)	54 (5.3)	<0.001 ***	52 (11.5)	<0.001 ***	148 (13.0)	56 (12.8)	0.929	22 (13.7)	0.810
**BMI, mean ± SD [median]**	28.1 ± 5.7 [27.3]	27.9 ± 5.6 [27.4]	0.540	27.4 ± 5.9 [26.7]	0.093	28.3 ± 5.8 [27.7]	26.4 ± 5.4 [25.7]	<0.001 ***	28.5 ± 5.1 [28.0]	0.674
**Extrahepatic metastases**	114 (28.1)	147 (14.5)	<0.001 ***	86 (19.0)	<0.01 **	227 (19.9)	70 (16.0)	0.077	28 (17.4)	0.451
**Liver condition-viral ^a^**	274 (67.7)	652 (64.2)	0.214	303 (66.9)	0.811	657 (57.6)	393 (89.9)	<0.001 ***	91 (56.5)	0.790
Hepatitis B virus	32 (7.9)	44 (4.3)	<0.01 **	20 (4.4)	<0.05 *	43 (3.8)	38 (8.7)	<0.001 ***	7 (4.3)	0.722
Hepatitis C virus	261 (64.4)	638 (62.8)	0.560	297 (65.6)	0.732	641 (56.2)	382 (87.4)	<0.001 ***	89 (55.3)	0.820
**Liver condition- non-viral**	48 (11.9)	130 (12.8)	0.628	54 (11.9)	0.975	171 (15.0)	13 (3.0)	<0.001 ***	31 (19.3)	0.163
**Cirrhosis**	296 (73.1)	710 (69.9)	0.230	314 (69.3)	0.224	802 (70.4)	313 (71.6)	0.619	113 (70.2)	0.966
**Child–Pugh score, n (%)**										
A (5–6 points)	345 (85.2)	681 (67.0)	<0.001 ***	348 (76.8)	<0.01 **	820 (71.9)	335 (76.7)	0.058	126 (78.3)	0.091
A5	220 (54.3)	356 (35.0)	<0.001 ***	214 (47.2)	<0.05 *	480 (42.1)	176 (40.3)	0.509	77 (47.8)	0.170
A6	125 (30.9)	325 (32.0)	0.681	134 (29.6)	0.683	340 (29.8)	159 (36.4)	<0.05 *	49 (30.4)	0.874
B (7–9 points)	56 (13.8)	319 (31.4)	<0.001 ***	101 (22.3)	<0.01 **	302 (26.5)	99 (22.7)	0.117	35 (21.7)	0.198
B7	37 (9.1)	199 (19.6)	<0.001 ***	70 (15.5)	<0.01 **	193 (16.9)	68 (15.6)	0.513	18 (11.2)	0.064
B8	15 (3.7)	94 (9.3)	<0.001 ***	22 (4.9)	0.407	84 (7.4)	26 (5.9)	0.322	11 (6.8)	0.807
B9	4 (1.0)	26 (2.6)	0.063	9 (2.0)	0.232	25 (2.2)	5 (1.1)	0.172	6 (3.7)	0.262
C (10–15 points)	4 (1.0)	16 (1.6)	0.396	4 (0.9)	1.000	18 (1.6)	3 (0.7)	0.166	0 (0.0)	0.152
C10	3 (0.7)	12 (1.2)	0.576	0 (0.0)	0.105	11 (1.0)	2 (0.5)	0.534	0 (0.0)	0.378
C11	1 (0.2)	4 (0.4)	1.000	4 (0.9)	0.377	7 (0.6)	1 (0.2)	0.457	0 (0.0)	1.000
**Ascites, n (%)**										
No evidence of ascites	380 (93.8)	863 (84.9)	<0.001 ***	411 (90.7)	0.091	986 (86.5)	406 (92.9)	<0.001 ***	144 (89.4)	0.300
Mild	21 (5.2)	130 (12.8)	<0.001 ***	37 (8.2)	0.082	134 (11.8)	24 (5.5)	<0.001 ***	15 (9.3)	0.363
Severe	4 (1.0)	23 (2.3)	0.112	5 (1.1)	1.000	20 (1.8)	7 (1.6)	0.834	2 (1.2)	1.000
**Encephalopathy, n (%)**										
No evidence of encephalopathy	404 (99.8)	1002 (98.6)	0.081	448 (98.9)	0.222	1130 (99.1)	432 (98.9)	0.575	158 (98.1)	0.211
Mild	1 (0.2)	13 (1.3)	0.131	5 (1.1)	0.222	9 (0.8)	5 (1.1)	0.550	3 (1.9)	0.177
Severe	0 (0.0)	1 (0.1)	1.000	0 (0.0)	1.000	1 (0.1)	0 (0.0)	1.000	0 (0.0)	1.000
**Modified ALBI grade, n (%)**										
Grade 1	138 (34.1)	213 (21.0)	<0.001 ***	143 (31.6)	0.435	306 (26.8)	112 (25.6)	0.625	42 (26.1)	0.839
Grade 2A	140 (34.6)	279 (27.5)	<0.01 **	130 (28.7)	0.065	326 (28.6)	132 (30.2)	0.529	55 (34.2)	0.146
Grade 2B	106 (26.2)	401 (39.5)	<0.001 ***	148 (32.7)	<0.05 *	391 (34.3)	155 (35.5)	0.662	56 (34.8)	0.904
Grade 3	21 (5.2)	123 (12.1)	<0.001 ***	32 (7.1)	0.254	117 (10.3)	38 (8.7)	0.349	8 (5.0)	<0.05 *
**Comorbidities**										
Diabetes mellitus	189 (46.7)	435 (42.8)	0.187	209 (46.1)	0.877	490 (43.0)	181 (41.4)	0.574	97 (60.2)	<0.001 ***
Hypertension	318 (78.5)	755 (74.3)	0.096	339 (74.8)	0.203	837 (73.4)	363 (83.1)	<0.001 ***	117 (72.7)	0.840
CCI, mean ± SD [median]	6.6 ± 2.6 [6.0]	5.8 ± 2.4 [5.0]	<0.001 ***	5.8 ± 2.4 [6.0]	<0.001 ***	6.0 ± 2.5 [6.0]	6.0 ± 2.5 [5.0]	0.596	6.0 ± 2.4 [6.0]	0.927

Abbreviations: A + B, atezolizumab + bevacizumab ALBI, albumin–bilirubin; BMI, body mass index; CCI, Charlson Comorbidity Index; CI, confidence interval; SD, standard deviation; uHCC, unresectable hepatocellular carcinoma; US, United States. Note: ^a^ Hepatitis B and C virus was assessed during the entire patient history (i.e., any time from data start [1 January 2017] to before the index date). * *p* < 0.05, ** *p* < 0.01, *** *p* < 0.001 when compared with atezolizumab + bevacizumab in the analysis by index treatment or with White patients in the analysis by race/ethnicity.

## Data Availability

Restrictions apply to the availability of these data. Data were obtained from the Veterans Health Administration National Corporate Data Warehouse and are available with the permission of Veterans Health Administration.

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
