# Peer review of "Overall Survival in Real-World Patients with Unresectable Hepatocellular Carcinoma Receiving Atezolizumab Plus Bevacizumab Versus Sorafenib or Lenvatinib as First-Line Therapy: Findings from the National Veterans Health Administration Database"

_cancers, 2024, doi:10.3390/cancers16203508_

Round 1

Reviewer 1 Report

Comments and Suggestions for Authors

The manuscript was well written and the topic is of interest. The authors should comment more on the potential impact of different etiologies on the final outcomes, as it was demonstrated that this parameter influences overall survival and progression-free survival in patients undergoing systemic treatments (for example lenvatinib: PMID: 36672330).

The authors should provide more content on the current state of the art on the topic (for example cite the recent SRMA: PMID: 34017396)

Please specify which R package did you use for the analysis. I guess the package "survival" was used here.

Author Response

Comment 1: The manuscript was well written and the topic is of interest.

Response 1: Thank you for your time reviewing our manuscript. We’re pleased that you found the topic of our study to be of interest to the readers of Cancers.

Comment 2: The authors should comment more on the potential impact of different etiologies on the final outcomes, as it was demonstrated that this parameter influences overall survival and progression-free survival in patients undergoing systemic treatments (for example lenvatinib: PMID: 36672330).

Response 2: Thank you for the suggestion. We have now added the following text to the Introduction section (lines 86-92) and cited Sacco et al. 2023:

“Due to the complex pathophysiology involved in HCC development, several efforts have been made to elucidate the impact of HCC etiology on treatment response and survival outcomes. For example, some prior real-world studies have observed an association between non-viral etiology and worse prognosis in patients treated with immune checkpoint inhibitors [14], likely due to a different hepatic microenvironment that consequently affects treatment response. However, the evidence for causality remains inconclusive [15].”

We added the following citation to the above text (#15):

- Meyer T, Galani S, Lopes A, Vogel A. Aetiology of liver disease and response to immune checkpoint inhibitors: An updated meta-analysis confirms benefit in those with non-viral liver disease. J Hepatol. 2023;79(2):e73-e6.

Comment 3: The authors should provide more content on the current state of the art on the topic (for example cite the recent SRMA: PMID: 34017396)

Response 3: Thank you for the suggestion. We have now cited the meta-analysis by Facciorusso et al. in the Introduction (lines 66-67):

“These TKIs were associated with similar OS and safety profiles as 1L therapy for uHCC [7].”

Comment 4: Please specify which R package did you use for the analysis. I guess the package "survival" was used here.

Response 4: Thank you for the question. We used the survival package and have now specified this in the Methods (line 184) as follows:

“Analyses were conducted using SAS 9.4 software (SAS Institute Inc., Cary, NC) and R software with the survival package (version 3.4.2; R Foundation for Statistical Computing).”

Reviewer 2 Report

Comments and Suggestions for Authors The publication entitled "Overall Survival in Real-World Patients with Unresectable Hepatocellular Carcinoma Receiving Atezolizumab Plus Bevacizumab versus Sorafenib or Lenvatinib as First-Line Therapy: Findings from the National Veterans Health Administration Database" by David E. Kaplan et al. focuses on evaluating the effectiveness of treating patients with unresectable hepatocellular carcinoma (uHCC) with the combination of atezolizumab and bevacizumab compared to standard first-line therapies (sorafenib and lenvatinib) in a real-world clinical setting in the United States. The authors aimed to: (1) Compare overall survival (OS) of uHCC patients treated with atezolizumab versus bevacizumab versus sorafenib and lenvatinib; (2) Investigate differences in treatment effectiveness by patient race and ethnicity and (3) Identify factors that influence differences in treatment outcomes among patients of different races and ethnicities. Advantages of the publication: (1) The article is based on a large sample of patients from real clinical settings, which strengthens its credibility; (2) The authors presented comprehensive statistical analyzes taking into account various factors such as race, ethnicity and liver function and (3) A significant contribution of the article is the analysis of racial and ethnic differences in the effectiveness of therapy, which may help to better understand differences in response to treatment.

Suggestions: 

  • More details could be added about the potential biological mechanisms behind racial and ethnic differences in treatment response.
  • Although the article raises important issues, some aspects, such as the limitations of the retrospective nature of the study, could have been discussed in more detail.
  • The writing style is formal and appropriate for a scientific publication, but sentences could be shortened in some places to increase readability.
  • All titles in the manuscript (headings and subheadings) are not prepared in accordance with the template and editorial requirements.
  • The way in which literature references are written is inconsistent with editorial requirements; instead of round brackets, e.g. (1), (2), (3), there should be square brackets [1], [2], [3], etc.
  • Table 1 has not been prepared in accordance with editorial requirements,
  • Figure 3 is illegible and has too small letters,
  • all References are prepared not in accordance with editorial requirements,
  • for all References, please provide all authors (do not use the abbreviation "et al."), e.g. Refs. 3, 7,9 and next,
  • when using websites (e.g. Ref. 2), please provide date of entry, as the information provided may change
  • Ref. 2, 8 and 30 please save them so that they do not constitute a link to the website (please remove the blue font and underline).

Summary: The reviewed article provides valuable data on the treatment of unresectable hepatocellular carcinoma in real-world clinical settings, taking into account racial and ethnic differences. The conclusions are important and may contribute to better adaptation of therapy to different groups of patients. The article is grammatically correct, although it could be enriched with further analysis of the potential causes of the observed differences in treatment outcomes. No significant errors were noticed that would affect the understanding of the content. The publication is based on 32 cited articles, mainly scientific ones, from the latest years. The subject of the work is interesting, but moderately innovative, because there are recent publications on the comparison of the effects of using the four drugs mentioned: atezolizumab and bevacizumab compared to standard first-line therapies (sorafenib and lenvatinib) in treating patients with unresectable hepatocellular carcinoma (uHCC), e.g.:

Journal of Liver Cancer 2024;24(1):81-91. DOI: https://doi.org/10.17998/jlc.2023.12.25

Therefore, I leave it to the Editorial Office to decide whether the publication can be published in such a prestigious journal as Cancers. My opinion is negative.

Author Response

Comment 1: The publication entitled “Overall Survival in Real-World Patients with Unresectable Hepatocellular Carcinoma Receiving Atezolizumab Plus Bevacizumab versus Sorafenib or Levatinib as First-line Therapy: Findings from the National Veterans Health Administration Database” by David E Kaplan et al. focuses on evaluating the effectiveness of treating patients with unresectable hepatocellular carcinoma (uHCC) with the combination of atezolizumab and bevacizumab compared to standard first-line therapies (sorafenib and levatinib) in a real-world clinical setting in the United States. The authors aimed to: (1) Compare overall survival (OS) of uHCC patents treated with atezolizumab and bevacizumab versus sorafenib and levatinib; (2) Investigate differences in treatment effectiveness by patient race and ethnicity; and (3) Identify factors that influence differences in treatment outcomes among patients of different races and ethnicities.

Advantages of publication: (1) The article is based on a large sample of patients from real clinical setting, which strengths its credibility; (2) The authors presented comprehensive statistical analyses taking into account various factors such as race and ethnicity, and liver function; and (3) A significant contribution of the article is the analysis of racial and ethnic differences in the effectiveness of therapy, which may help to better understand differences in response to treatment.

Response 1: Thank you for your time reviewing our article and for your helpful suggestions on improving the article. We’re pleased that you found the analyses to be comprehensive and that our focus on the impact of race/ethnicity on treatment effectiveness was a significant contribution with potential clinical impact for patients.

Comment 2: Suggestions:

More details could be added about the potential biological mechanisms behind race and ethic differences in treatment response.

Response 2: Thank you for the suggestion. On page 15 in the Discussion, we stated, “Racial and ethnic variations in treatment response to immunotherapy, including atezolizumab plus bevacizumab, may arise from differences in genetic predispositions, access to care, or underlying disease etiologies.”

We have now elaborated on this to add (lines 356-366):

“Both race and geographical location are independent risk factors for HCC, while other risk factors (i.e., obesity, diabetes, hepatitis B and C, metabolic dysfunction associated steatotic liver disease or steatohepatitis, smoking, etc.) are differentially prevalent among racial and ethnic subgroups [31]. Additionally, chronic stress related to poverty and/or discrimination, as well as lack of health insurance or difficulty accessing care may also impact cancer prognosis and treatment response, and timeliness of treatment initiation [31-33]. Further, there are racial and ethnic differences in health literacy and medical mistrust, which can impact engagement with medical care and adherence to treatment regimens [34]. Finally, racial differences in OS have been partially attributed to treatment delay and variation in utilization of HCC treatment [33, 35].

We added the following citations to the above text:

- Mathur AK, Osborne NH, Lynch RJ, Ghaferi AA, Dimick JB, Sonnenday CJ. Racial/ethnic disparities in access to care and survival for patients with early-stage hepatocellular carcinoma. Arch Surgery. 2010;145(12):1158-63.

- Thylur RP, Roy SK, Shrivastava A, LaVeist TA, Shankar S, Srivastava RK. Assessment of risk factors, and racial and ethnic differences in hepatocellular carcinoma. JGH Open. 2020;4(3):351-9.

- Wagle NS, Park S, Washburn D, Ohsfeldt RL, Rich NE, Singal AG, Kum H-C. Racial, ethnic, and socioeconomic disparities in treatment delay among patients with hepatocellular carcinoma in the United States. Clin Gastroenterol Hepatol. 2023;21(5):1281-92.e10.

- Wong RJ, Corley DA. Survival differences by race/ethnicity and treatment for localized hepatocellular carcinoma within the United States. Dig Dis Sci. 2009;54(9):2031-9.

- Schoenberger H, Rich NE, Jones P, Yekkaluri S, Yopp A, Singal AG. Racial and ethnic disparities in barriers to care in patients with hepatocellular carcinoma. Clin Gastroenterol Hepatol. 2023;21(4):1094-6.e2.

Comment 3: Although the article raises important issues, some aspects, such as the limitations of the retrospective nature of the study, could have been discussed in more detail.

Response 3: Thank you for the comment. We discussed the limitations of the retrospective nature of the study, but have now added the following text to elaborate (lines 403-405):

“Third, there is the potential for bias with regards to the classification for key variables such as etiology identified through diagnosis code and Child-Pugh class from a derived algorithm.”

Comment 4: The writing style is formal and appropriate for a scientific publication, but sentences could be shortened in some places to increase readability.

Response 4: Thank you for the comment. We have now simplified the language structure in the manuscript when feasible.

Comment 5: All titles in the manuscript (heading and subheadings) are not in proper accordance with the template and editorial requirements. The way in which literature references are written are inconsistent with editorial requirements; instead of round brackets, e.g.. (1), (2), (3), there should be square brackets [1], [2], [3], etc. Table 1 has not been prepared in accordance with editorial requirements.

Response 5: Thank you for the comment. Cancers indicates in their online instructions to authors that they accept format-free submissions. The manuscript text has now been placed into the journal template (with the formatted headings and subheadings) and our edits have been enacted in that version. Additionally, the citation and table formats have been updated to the journal style.

Comment 6: Figure 3 is illegible and has too small letters.

Response 6: Thank you for the comment. We have now increased the text size in Figure 3.

Comment 7: All references are prepared not in accordance with editorial requirements. For all references, please provide all authors (do not use the abbreviation “et al.”), e.g., Refs 3, 7, 9, and next, when using websites (e.g., Ref 2), please provide the date of entry, as the information provided may change. Ref 2, 8, and 30 please save them so they do not constitute a link to the website (please remove the blue font and underline).

Response 7: Thank you for the comment. Similar to our above response regarding the formatting, all references have now been formatted for the journal style with all authors listed. For website citations, we have now included the access dates and saved urls as non-hyperlinked.

Comment 8: Summary: The reviewed article provides valuable data on the treatment of unresectable hepatocellular carcinoma in real-world clinical settings, taking into account racial and ethnic differences. The conclusions are important and may contribute to better adaptation of therapy to different groups of patients. The article is grammatically correct, although it could be enriched with further analysis of the potential causes of the observed differences in treatment outcomes. No significant errors were noticed that would affect the understanding of the content. The publication is based on 32 cited articles, mainly scientific ones, from the latest years.

Response 8: Thank you for your insight into our study. It’s gratifying to hear that you believe the study conclusions are important, and it is also our hope that the results can help optimize therapy for patients with uHCC across demographic groups. As mentioned in our response above, we have now added more discussion surrounding the potential causes of the observed differences in treatment outcomes.

Comment 9: The subject of the work is interesting, but moderately innovative, because there are recent publications on the comparison of effects of using the four drugs mentioned: atezolizumab and bevacizumab compared to standard first-line therapies (sorafenib and levatinib) in treating patients with unresectable hepatocellular carcinoma (uHCC), e.g., Journal of Liver Cancer 2024;24(1):81-91. DOI: https://doi.org/10.17998/jlc.2023.12.25. Therefore, I leave it to the Editorial Office to decide whether the publication can be published in such a prestigious journal as Cancers. My opinion is negative.

Response 9: Thank you for the comments. Regarding the article you noted by Park et al., our study fundamentally differs in multiple ways including the patient population’s characteristics, the sample size, and therapies compared. All patients in Park et al. were Korean and had portal vein tumor thrombosis (PVTT) in addition to uHCC, so their results are applicable to a more specialized patient population outside of the US. As you have noted in your summaries, a strength of our study is the analysis of the impact of race and ethnicity on treatment outcomes among a diverse patient population. The current study utilized data from the National Veteran Health Administration (VHA) database – the largest healthcare system providing liver care in the US, generating results most relevant to the HCC population. Additionally, Park et al. had a small sample size of only 52 patients and short follow-up time (median 6.4 months) while 1,874 patients were included in our analyses with a longer median follow-up of up to 8.5 months. Finally, Park et al. compared OS only between atezolizumab plus bevacizumab vs. lenvatinib, while we also included comparison with sorafenib. Thus, the results of the two studies are complementary towards expanding the literature on the outcomes of patients with uHCC, but do not overlap.

We appreciate your thorough review and hope that our responses to your comments and updates to the manuscript have helped to satisfy your concerns about meriting publication in Cancers.

Reviewer 3 Report

Comments and Suggestions for Authors

Manuscript ID: cancers-3205416

Title: Overall Survival in Real-World Patients with Unresectable Hepatocellular Carcinoma Receiving Atezolizumab Plus Bevacizumab versus Sorafenib or Lenvatinib as First-Line Therapy: Findings from the National Veterans Health Administration Database

Review Summary:

This is a well-written and well-organized manuscript describing a retrospective analysis of overall survival between different pharmacologic therapies for first-line treatment of unresectable hepatocellular carcinoma. Patient data were obtained from VHA electronic health records over a well-defined period. Patient demographics and clinical characteristics were included in the statistical analyses to probe possible important covariates influencing OS for the three evaluated therapies. The conclusions are strong and would appear to this reviewer to be clinically significant to physicians and patients. Assumptions and limitations of the study are presented.

Major comments: None

Minor comments:

In the Introduction, please include the mechanisms by which atezolizumab (PD L1) and bevacizumab (VEGF) are effective. Mechanism is included for the two TKIs on Line 48 of the Introduction.

Table 1 has [median] and [IQR] listed in row for BMI, yet only one bracketed value is shown. Please clarify.

Methods define OS (Lines 122-124); however, Results present unadjusted OS. Please clarify how unadjusted OS was calculated.

Methods: Please define how Probability of Survival was calculated.

Author Response

Comment 1: This is a well-written and well-organized manuscript describing a retrospective analysis of overall survival between different pharmacologic therapy for first-line treatment of unresectable hepatocellular carcinoma. Patient data were obtained from VHA electronic health records over a well-defined period. Patient demographics and clinical characteristics were included in the statistical analyses to probe possible important covariates influencing OS for the three evaluated therapies. The conclusions are strong and would appear to this reviewer to be clinically significant to physicians and patients. Assumptions and limitations of the study are presented.

Response 1: Thank you for your time reviewing our article and for your insightful comments. We’re pleased that you felt that the conclusions have clinically significant implications for physicians and patients.

Comment 2: Minor comments:

In the Introduction, please include the mechanisms by which atezolizumab (PD L1) and bevacizumab (VEGF) are effective. Mechanism is included for the two TKIs on Line 48 of the Introduction.

Response 2: Thank you for the suggestion. We have now listed the mechanisms for these two therapies as you recommend (lines 71-77):

“Atezolizumab, an immune checkpoint inhibitor, is a monoclonal antibody which binds to programmed death ligand 1 (PDL1) and blocks its interaction with PD1 and B7.1 receptors [8]. Bevacizumab is a humanized monoclonal IgG1 antibody which binds to vascular endothelial growth factor (VEGF) ligand and has anti-angiogenic and immuno-modulatory effects [9]. Clinical and preclinical studies demonstrated that anti-PD-L1 plus anti-VEGF lead to a significant improvement in survival and enhanced anti-tumor activity [10].”

We cited:

- Casak SJ, Donoghue M, Fashoyin-Aje L, Jiang X, Rodriguez L, Shen Y-L, et al. FDA approval summary: Atezolizumab plus bevacizumab for the treatment of patients with advanced unresectable or metastatic hepatocellular carcinoma. Clin Cancer Res. 2021;27(7):1836-41.

- Garcia J, Hurwitz HI, Sandler AB, Miles D, Coleman RL, Deurloo R, et al. Bevacizumab (Avastin®) in cancer treatment: A review of 15 years of clinical experience and future outlook. Cancer Treat Rev. 2020;86:102017.

- Kudo M. Scientific rationale for combined immunotherapy with PD-1/PD-L1 antibodies and VEGF inhibitors in advanced hepatocellular carcinoma. Cancers. 2020;12(5):1089.

Comment 3: Table 1 has [median] and [IQR] listed in row for BMI, yet only one bracketed value is shown. Please clarify.

Response 3: Thank you for noting this. We removed the bracket for IQR. The bracket shown is the median.

Comment 4: Methods define OS (Lines 122-124); however, Results present unadjusted OS. Please clarify how unadjusted OS was calculated. Please define how Probability of Survival was calculated.

Response 4: Thank you the suggestion. We have now described how unadjusted OS was calculated in the Methods section, as follows (lines 161-163):

“Unadjusted OS was calculated using the Kaplan-Meier (KM) method and the probability of survival at a given time point was derived directly from the KM survival curve.”

Round 2

Reviewer 1 Report

Comments and Suggestions for Authors

The manuscript is OK in the current form. Thank you!

Reviewer 2 Report

Comments and Suggestions for Authors Thank you for all your answers
and for taking into account all my comments and suggestions.
After introducing the suggested corrections, the manuscript gained in value.
If the Editors agree, your manuscript may be published.